# An Innovative Auxetic Honeycomb Sandwich Tube: Fabrication and Mechanical Properties

**DOI:** 10.3390/polym14204369

**Published:** 2022-10-17

**Authors:** Jianqin Wu, Jiannan Zhou, Xinli Kong, Ying Xu, Yishun Chen, Juyan Zhu, Fengnian Jin, Peng Wang

**Affiliations:** 1State Key Laboratory of Disaster Prevention & Mitigation of Explosion & Impact, Army Engineering University of PLA, Nanjing 210007, China; 2JiangSu Cicil Air Defence Works Quality Supervision Station, Nanjing 210036, China

**Keywords:** auxetic honeycomb, sandwich structures, Poisson’s ratio, failure mode

## Abstract

In this study, based on the free-rolling mechanism of the auxetic honeycomb, a honeycomb cylindrical shell was successfully prepared to overcome the fracture problem of the hexagonal honeycomb during rolling. Auxetic honeycomb sandwich tubes (AHSTs) with a variable Poisson’s ratio were fabricated by molding and bonding. A Poisson’s ratio model of the auxetic honeycomb core was developed based on the strain increment ratio of the deformed honeycomb and validated using computed tomography (CT). Four failure modes (progressive stable fold mode I, unstable local buckling mode II, transverse shearing mode III, and mid-length collapse mode IV) of the AHST were summarized by comparing the deformation behavior and force–displacement curves with different geometric parameters. When the aspect ratio R is greater than 3, the AHST will be more easily damaged in instability (Mode IV). Static compression tests showed that the peak force (PF) and crushing force efficiency (CFE) of the AHST were higher than those of the CFRP thin-walled tube of the same diameter by 78% and 115%, respectively. Therefore, the AHST has excellent mechanical properties and it is feasible to use the auxetic honeycomb as a core for sandwich structures.

## 1. Introduction

Composite sandwich structures are widely used in the aerospace, marine, and automotive industries for their high flexural stiffness-to-weight ratio and excellent energy absorption abilities [1,2,3]. Composite thin-walled structures have excellent designability, while honeycomb sandwich composites have great topology design potential, and the combination of composite thin-walled structures and composite honeycomb cores introduces greater flexibility and more design space to meet the requirements of variable design [4].

Composite thin-walled structures are used to sandwich structures due to their light weight, high-strength material properties, and flexible designability. Mamalis et al. [5,6] conducted an experimental study on the damage mechanism of CFRP tubes under axial compressive loading. The experimental results showed that all tested CFRP tubes collapsed in a brittle manner under compressive loading due to the brittle characteristics of the materials consisting of carbon-fiber-reinforced fibers and epoxy resin. Three brittle damage modes were observed: progressive end crush mode I, localized tube wall buckling mode II, and medium-length damage mode III. However, the use of a single composite material has many shortcomings, such as brittle damage under high-strain loads and expensive fabrication costs, and the mechanical properties of the structure are greatly influenced by environmental factors.

Hugo Junkers first proposed the idea of using a honeycomb core between two panels in 1915. In recent years, researchers have tried to use porous materials (foam, honeycomb, lattice, etc. [7,8,9,10,11]) as cores in sandwich structures filled with composite thin-walled structures. This way has proven to be effective to improve their energy absorption capacity [12,13]. By far the most widely used honeycomb structure is the hexagonal honeycomb structure, but other honeycomb structures (Kagome honeycomb, auxetic honeycomb, triangular, square, and circular) have properties (negative Poisson’s ratio, thermal conductivity, higher stiffness, etc.) that the hexagonal honeycomb does not have.

The frequently used honeycomb has a hexagonal honeycomb shape, which is easy to manufacture and ideal for making flat sandwich panels. One disadvantage of a hexagonal honeycomb is that if it is bent out of the plane, it will create a saddle-shaped curve due to the positive Poisson’s ratio [14]. As shown in Figure 1, for the honeycomb double-curved structure, local fragmentation of the honeycomb will be caused by forcing the honeycomb panels into the desired shape. However, if the effective Poisson’s ratio is made negative by changing the cell shape, the centripetal curvature in the dome plane can be achieved for the negative Poisson’s ratio. For honeycombs subjected to in-plane loads, relatively low initial forces are more suitable for quasi-static compression. In addition, the complex collapse mechanism of the in-plane honeycomb enables the honeycomb topology to significantly affect the honeycomb crushing behavior and further tune the Poisson’s ratio of the honeycomb core.

The designability of the auxetic honeycomb allows for directional design. Various forms are possible such as the reentrant honeycomb [15], arrowheads [16], chirals, and more complex bio-inspired forms [17]. The auxetic honeycomb structure has been designed and applied to sensors [18], energy-absorbing structures, anti-explosion structures [19], flexible electronics [20], and many other fields, due to its excellent shear strength, enhanced fracture toughness, and obvious indentation resistance [21]. The auxetic honeycomb structure is porous, and it is meaningful to study the characterization of its auxetic honeycomb structure [22].

In terms of theoretical research, Master and Evans [23] calculated the elastic constants of the honeycomb, such as tensile modulus, shear modulus, and Poisson’s ratio, by considering the deformation of auxetic units. Yang et al. [24] predicted the modulus, Poisson’s ratio, and yield strength of the auxetic honeycomb based on the Timoshenko beam model. Berinskii [25] proposed a deterministic model for the effective elastic characteristics of the auxetic honeycomb plane, and the change in the angle of the honeycomb unit triggered the structure to switch from auxetic to traditional, causing a qualitative change in the mechanical properties of the honeycomb. These mechanical properties are very sensitive to the specific range of the geometric parameters of the honeycomb units.

In this composite sandwich structure, the main energy absorption is due to the deformation of the skin and sandwich structure, so the failure mechanism of the core and skin is the key factor of the energy absorption capacity of the structure, but there are still relatively few studies in this part. 

Based on the free-rolling mechanism of the honeycomb, a fabrication process for auxetic honeycomb sandwich tubes (AHSTs) is proposed in this paper. The CFRP skin is processed by hand lay-up and the AHST is manufactured via molding and bonding, and the whole fabrication process is easy and feasible with low cost. As shown in Figure 2, the AHST consists of a CFRP outer skin, auxetic honeycomb core, and CFRP inner skin. In this paper, the uniaxial compression Poisson’s ratio analysis model of the honeycomb structure is established, and the change in Poisson’s ratio of the auxetic honeycomb after rolling is studied and predicted based on plastic large deformation theory. The failure modes of sandwich tubes with different sandwich structures and different aspect ratios were obtained by axial compression tests, and the mechanical properties and energy absorption characteristics of auxetic honeycomb sandwich tubes were investigated.

## 2. Design and Fabrication

### 2.1. Design of the Honeycomb Core

A method for fabricating honeycomb cores with a variable Poisson’s ratio for the sandwich tubes is presented in this section. As shown in Figure 3, the sandwich structure is formed by rolling the honeycomb plate along the armchair direction into a cylindrical shape. The complete core is obtained by arraying m cells in the circumferential direction and n cells in the axial direction. The angle of each cell to the center is *α*, which can be calculated by *2π/m*. The thickness of the honeycomb core is donated as *b*. Both the inner and outer faces have a thickness of *t_s_*. The outer diameter of the tube is *D* and the inner diameter is d. The height of the tube structure is *H*. The size of each unit is shown in Figure 3d. The length of the circumferential arc is *h*, the length of vertical ribs is *l*, the reentrant angle is *θ*, and the thickness of the unit wall is *t*.

The Poisson’s ratio of hexagonal honeycombs is directly affected by the cell topology and can usually be divided into reentrant and convex honeycombs. Through this topological transformation, as shown in Figure 4, the Poisson’s ratio of the cells also changes from negative to positive (convex cells with positive Poisson’s ratio, with zero Poisson’s ratio, and with negative Poisson’s ratio, and reentrant cells). As shown in Figure 4, the reentrant angle *θ* of honeycomb units is diverse due to the geometric deformation, which also causes the Poisson’s ratio of the structure to vary. 

Because the honeycomb panel is undeformed before rolling, assuming that such a face still exists in the core, the length of the face is
(1)$$C=2hm=d_{0}\pi$$
where *d_0_* is the diameter of the middle face. In this formula, when the number and size of the honeycomb unit are determined, the middle face is also determined.
(2)$$d_{0}+2\Delta x=d_{1}$$
(3)$$d_{1}\;\pi =2m\left(h-l\;cos\theta \right)$$
(4)$$arc\;cos\left(\frac{\Delta x\pi }{ml}\right)=arc\;cos\left(\frac{\alpha \Delta x}{2l}\right)=\theta ,\theta \in \left[\frac{\pi }{3},\pi \right]$$
where *d_1_* is the diameter of the deformed face. When the middle face moves Δ*x* along the *x*-axis, the honeycomb unit deforms accordingly, as shown in Figure 4. However, the deformation of each honeycomb unit varies in the radial direction, and the relationship between *θ* and *x* can be observed through the geometry. This interesting phenomenon is beneficial to design a sandwich tube with a variable Poisson’s ratio. It is worth noting that when the angle *θ* is equal to *π/3*, the unit becomes a triangular unit when *h* is equal to *l*, and the minimum radius of the tube is *d_min_* (the diameter of the inner face).
(5)$$\;d_{min}=\frac{ml}{2\pi }$$

Here, a plastic, large deformation analysis is performed for the unit cell. Assume that the cell wall material is rigid/perfectly plastic, without any elastic deformation. The strain can be calculated from the displacements corresponding to the deformation mechanism. *H_h_* and *L_h_* are the current height and width of the honeycomb unit cell, respectively, during the deformation. *H_0_* and *L_0_* are the height and width before deformation, respectively.
(6)$$\;H_{0}=2l\;\;\;\;\;\;\;\;\;\;\;\;\;\;\;\;\;\;\;\;\;\;\;\;\;\;\;\;\;\;\;\;\;\;\;\;\;\;\;\;L_{0}=2h$$
(7)$$\;\;\;\;H_{h}=2l\sin\theta \;\;\;\;\;\;\;\;\;\;\;\;\;\;\;\;\;\;\;\;\;\;\;\;\;\;\;\;L_{h}=2h-2l\cos\theta \;$$

The engineering strain can be calculated as [24]
(8)$$\;\varepsilon_{1}=\frac{H_{h}-H_{0}}{H_{0}}=\sin\theta -1\;\;\;\;\;\;\;\;\;\;\;\;\;\;\;\;\;\;\varepsilon_{2}=\frac{L_{h}-L_{0}}{L_{0}}=-\frac{l}{h}\cos\theta$$

Consequently, the Poisson’s ratio can be calculated as [25]
(9)$$\;v_{1}=-\frac{\varepsilon_{2}}{\varepsilon_{1}}=\frac{l}{h}\frac{\cos\theta }{\sin\theta -1}\;\;\;\;\;\;\;\;\;\;\;\;\;\;\;\;v_{2}=-\frac{\varepsilon_{1}}{\varepsilon_{2}}=\frac{h}{l}\frac{\sin\theta -1}{\cos\theta }$$

Based on this configuration, the core with a variable Poisson’s ratio is designed. According to Equation (9), the relationship between geometry and Poisson’s ratio is shown in Figure 5. From a topological point of view, Poisson’s ratio is zero when *θ* is 90°. When the reentrant angle *θ* is greater than 90°, the absolute value of Poisson’s ratio increases as the angle decreases. When it is less than 90°, the absolute value of Poisson’s ratio increases with the reentrant angle decrease. This tends to produce larger transverse strains with longitudinal strains when the reentrant angle *θ* is close to 90°, thus producing a higher absolute value of Poisson’s ratio. The zero Poisson’s ratio is independent of geometric dimensions (*h* and *l*) and is only determined by the reentrant angle *θ*. According to the upper and lower plastic boundaries [26], the static strength *σ_0_* is directly written as:(10)$$\;\sigma_{0}=\frac{\sigma_{y}\;t^{2}}{\left(2l\left(h-lsin\theta \right)sin\theta \right)}$$

### 2.2. Verification of the Honeycomb Angle Variation

Industrial CT, the computer tomography technology, is employed to verify the reentrant angles of the deformed honeycombs. By rotating the tested specimens and reconstructing the 2D projection data collected in different directions, the 3D CT results of the tested specimens, that is, all of the geometric dimensions and material information of the tested specimens, can be obtained. The graphic of honeycombs is acquired with an accelerating voltage of 90 kV and beam current of 90 μA. The AHST is scanned for 2000 ms and a series of honeycomb unfolding diagrams with a resolution of 2.5 μm are obtained, as shown in Figure 6.

The scanned systems including the ‘Diondo d2 CT system’ uses the industry’s micro- and nano-focus ray sources, with micro-focus sources up to 300 kV for high-density, small-size specimens. For high-resolution CT scanning, additional nano-focus sources are available with a maximum spatial resolution of 0.5 mm, while image quality can be obtained due to high-contrast, high-dynamic-range flat panel detectors with a wide range of product quality control, scientific research, and analysis requirements.

The geometric parameters of the tubes used in this experiment are *m* = 34, *D* = 60 mm, *h* = *l* = 2.75 mm, and Δ*x* = *b* = 10 mm. The compressive strength of the honeycomb is 2.21 MPa, the shear strength is 1.28 MPa, and the elastic modulus is 0.117 GPa. Combined with Equations (1)–(4), the value of *θ* was calculated as 71.4°. According to Equation (9), the Poison’s ratio can be obtained as −5.6, and the strength can also be obtained by Equation (12), *σ_0_* = 0.27 MPa.

As shown in Figure 7, the measured angles in the inner face are very close to the calculated result, and the correctness of the formulas can be verified.

### 2.3. Material Property

The CFRP tubular structure is made of a single-layer carbon-fiber braid by the conventional process of one-over and one-under, as shown in Figure 8, with one fabric with 144 bundles and 72 tows in each direction at different angles. The fibers are all composed of 3k carbon fiber bundles, and the linear density of each carbon fiber bundle is 0.2 g/m, so the linear density of 144 carbon fiber bundles is 28.8 g/m. Epoxy resin is used as the resin base material, its density is 0.93 g/cm^3^, its compressive strength is 91 MPa, its tensile strength is 57 MPa, its elastic modulus is 2.69 GPa, and its elongation is 2.66%. 

The aramid paper honeycomb begins life as a coil of foil. Foil goes through the printer for adhesive lines to be printed, in Figure 9. Then, the foil is cut to size and stacked into piles using the stacking machine. Following this, the stacked sheets of foil are pressed using a heated press to allow the adhesive to cure and bond the sheets of foil together, forming a block. The block can be cut into slices. Finally, the honeycomb is expanded. The expanding degree determines the topology of the honeycomb [14]. The density of the honeycomb used in this study is 48 kg/m^3^, and it is cut into a predetermined rectangle for later use. In this study, the wall thickness *t_s_* of the CFRP tube is 0.5 mm, *h* and *l* of the cell walls length are 2.75 mm, the wall thickness *t* of the honeycomb unit is 0.2 mm, and the radial width *b* of the honeycomb unit is 10 mm. In addition, the material properties of epoxy resin and honeycomb are summarized in Table 1.

### 2.4. Fabrication

As shown in Figure 10, the AHST is made in four steps. First, the carbon fiber composite tube is made by hand brushing epoxy resin onto carbon fiber and soaking the epoxy resin between the fibers. Secondly, the rolling honeycomb core is put into the carbon fiber tube to make the sandwich structure. Third, the production process in the first step is used to make a carbon fiber tube for the inner face. Finally, the carbon fiber face, honeycomb core, and carbon fiber inner face are bonded together to form the AHST.

## 3. Experiments and Analyses

### 3.1. Experimental Schemes

A total of six types of AHSTs were made, four types of which have only the outer face, and the other two types have both inner and outer faces. Two different tube diameters *D* and three tube lengths *H* were considered to form a four-aspect ratio (the ratio of the tube length to tube outer diameter *R*). AHSTs with different aspect ratios were used to observe failure modes during the test.

As listed in Table 2, the geometric dimensions and mass of the specimens are summarized. The tubes are labeled according to the material type, sandwich structure, tube length, and diameter. For example, specimen 60D40-CH has a tube length of 60 mm and a tube outer diameter of 40 mm. The sandwich structure has only the outer face, and the honeycomb sandwich can be constrained by the CFRP tube. The 60D60-CHC means that the tube length is 60 mm, the tube outer diameter is 60 mm, and the sandwich structure has both inner and outer faces, where the inner CFRP tube can improve the overall stability of the sandwich structure.

The quasi-static axial compression tests were carried out on the universal testing machine, with a load capacity of 150 kN. Throughout the test, the loading speed remained unchanged at 2 mm/min. Generally, the effective crushing distance (ECD) of the CFRP tube (the ratio of the crushing distance to the tube length) was about 80%, so the end position of the compression displacement was set at 80% of the tube length. The system recorded the force–displacement curve of the tested specimens.

### 3.2. Axial Compression Performances

#### 3.2.1. Macro-Failure Modes and Crushing Histories

The failure modes of the sandwich tube of auxetic honeycombs were related to their sandwich structure and slenderness. Based on previous studies [29,30,31,32,33,34] and the experimental results of this study, four macro-failure modes of the AHST were determined, namely Mode I—progressive stable fold mode, Mode II—unstable local buckling mode, Mode III—transverse shearing mode, and Mode IV—mid-length collapse mode. The crushing snapshots of the AHST show the deformation modes at the different strains of the sandwich tube specimens. Ε_1_ is the strain corresponding to the initial peak load, ε_2_ is 0.4, ε_3_ is 0.6, and ε_4_ is 0.8.

Generally, the compressive force–displacement curve of the composite tube can be divided into two stages, namely the elastic deformation stage and the progressive deformation stage. In the elastic stage, the crushing load increased rapidly and reached a peak soon after. At the same time, the tube tended to collapse and the crushing load suddenly dropped to a lower level. Since then, the tube turned into the progressive deformation stage, where the crushing load fluctuated in a small range around the mean force. However, the progressive deformation stage was divided into some stages, because multiple failure modes appeared simultaneously in some special cases. In Figure 11, Figure 12 and Figure 13, the solid line represents the force and the dashed line represents the energy absorbed by the tubes during compression.

It can be seen that the failure of 120D60-CHC was in Mode I with the oblique shear band in the tube wall (Figure 11). Due to the circumferential shearing effect caused by local stress concentration on the shear band, the ring-shaped failure surfaces were folded at the band. The crushing behaviors of the whole model were somewhat similar to progressive plastic deformations of thin-walled metal tubes, where many plastic hinges were developed. 

The failure of 120D60-CH was in Mode II with local instability and deformation in the tube, which hindered the further development of the buckling zone. As a result, irregular brittle fractures occurred around the buckling paths. It is worth noting that a sharp groove on the tube wall was formed and accompanied by fiber fracturing in the unstable buckling zone.

The crushing behavior of 120D60-CH was mainly due to the unstable local buckling and CFRP folds, resulting in the fluctuation in load in the force–displacement curve. For the 120D60-CHC, the load fluctuated in a smaller range around the mean force owing to forming a large number of regular folds during the crushing. In addition, the mean forces of 120D60-CHC and 120D60-CH were different due to their respective deformation modes. The 120D60-CH tube retained a lot of undamaged zones stemming from unstable local deformation.

The crushing histories of the 60D60-CHC tube are shown in Figure 12. The failure of 60D60-CHC was in Mode I, where oblique shear bands were formed in the tube wall and the ring-shaped failure surfaces were folded on the band. Many small regular folds were developed during the progressive stable deformation in the tube wall. 

The failure mode of the 60D60-CH tube was a combination of Mode I and Mode II, in which Mode II dominated the deformation in the first half of the crushing process, then Mode I played a major role in the second half of the crushing process. In the first stage (1–2), the path of buckling was an irregular curve instead of a straight band. In this case, the unstable local bucking deformation was replaced by the progressive stable deformation, and fiber fracturing took place at the second stage (2–4) of the crushing process.

As for 60D60-CHC and 60D60-CH, Mode I was characterized by stable progressive local buckling folds, leading to a series of small oscillations around the mean force, which was fairly similar to that of its metal counterpart. Interestingly, the mean crushing forces of the 60D60-CHC tubes were larger than those of the 60D60-CH tubes. This was because the unstable local buckling deformation occurred in 60D60-CH.

As shown in Figure 13, the progressive stable fold and transverse shearing were observed in the 60D40-CH tube, and the failure appeared to be a combination of Mode I and Mode III. The failure of the first stage (1–2) was in Mode III, in which the slope of the tube wall was formed due to the tube transverse shearing. At this time, the tube was under eccentric compression. Due to the tube having a larger diameter-to-thickness ratio (the ratio of the outer diameter of the tube to the thickness of the sandwich) and smaller slenderness, Mode I occurred in the next stage (2–3).

With the increase in the height of the tube, the collapse modes of the 120D40-CH tubes were quite different from the other tubes, the failure was in Mode IV, and the tube collapsed in the middle and was divided into two slope parts.

As for 60C40-CH, Mode III failure dominated the deformation in the first half of crushing, and the force–displacement curve dropped abruptly to a lower level because of transverse shearing. Then, Mode I failure appeared in the second half of the crushing process, and there was a series of moderate oscillations around the mean force on the force–displacement curve. The peak load of 60D40-CH was higher than that of 120D40-CH, due to the crushing mode of the latter being the mid-collapse mode, which made the load suddenly drop to a low level and fluctuate to a small value.

#### 3.2.2. Failure Mechanisms

The failure mode of the sandwich structure was the result of the competition between the face and the core. Therefore, it is necessary to study the damage to the face, the core, and their interface. Based on this experiment and previous studies, the auxetic honeycomb core could be divided into three micro-failure modes: ring mode, Z mode, and mixed mode, as shown in Figure 14. The three main failure processes were accompanied by failure phenomena such as face wrinkling, intra-cell buckling, face sheet crushing, core member crushing, and debonding.

Figure 14a shows the ring mode failure of one of the 60D60-CH specimens, and the honeycomb core presented a significant negative Poisson’s ratio, which was caused by the cell shrinking. The radius of the middle area of the concave structure was reduced, leading to the interface debonding. As the crushing progressed, there were many shear bands formed on the surface of structures, accompanied by the twisting of cells (Figure 14d). Then, the cells had a high degree of compactness and stably formed a dense layer. It can be seen from the micro-failure modes that the oblique shear band in Mode I was caused by the twisting of cells. In the ring mode, the tube wall was folded progressively, but due to the negative Poisson’s ratio effect, the size of the folded cells in the structure was inconsistent.

As shown in Figure 14b, in this crushing process of one of the 120D60-CH specimens, the response of the CFRP face was controlled by the combination of collapsing of the honeycomb core and debonding of the interface. In addition, core debonding occurred as the bonding layers were not strong enough to transmit the constraints of the honeycomb cores. The cells had a low degree of compactness due to the cells not only having the in-plane deformation but also having unstable out-of-plane deformation during the buckling of the face. According to the previous analyses of macro-failure modes, the Z mode stemmed from Mode II failure, in which unstable local buckling dominated the failure of the crushing process.

Figure 14c shows the mixed failure mode of one of the 120D60-CHC specimens; the left side of the section is the ring mode failure and the right side is the Z mode failure. At the same time, the failure mode presented all the above-mentioned failure phenomena. In the ring mode failure side, the face was folded, the core was contracted and concaved, and the cells were folded inwardly and densely. The deformation of the structure was mainly controlled by the CFRP face, and the local buckling zone of the CFRP face continued to appear until the tube was densified. In the Z mode failure side, the face was folded inward, and the honeycomb core was collapsed out-of-plane. It belonged to the mixed macro-failure modes of Mode I and Mode II. 

When there was only a single outer face, part of the auxetic cells suffered from buckling first and then transformed into rhomboid cells. In this process, due to the twisting of the cells, the structure appeared to expand and contract inward. This deformation mode converted the structure into an unstable situation, resulting in the crushing of the core at the inner fold and twisting during the subsequent crushing process. When there were double faces, the deformation of the face was constrained by the core; therefore progressive buckling formed stably, and the two faces could restrain the twisting of the core. A negative Poisson’s ratio appeared in the crushing process, but the tube was not rotating, due to the restraining of double faces.

#### 3.2.3. Crashworthiness Indicators

*PF* (peak force), *MCF* (meaning crashing force), *SEA* (specific energy absorption), and *CFE* (crushing force efficiency) were chosen as evaluation indicators of *EA* (energy absorption) performance, as shown in Table 3.

*SEA* (specific energy absorption) is the energy absorption by the specimens per unit mass, which presents the utilization efficiency of specimens.
(11) SEA=EA/W
where *W* is the mass of specimens, *EA* is the energy dissipated in the process of loading, and
(12)$$EA=\int_{0}^{ECD}F\left(x\right)\;dx$$
where *ECD* is the effective compression distance, which is shown in Table 2, and represents the corresponding displacement when the force in the force–displacement curve is equal to the peak force at the stage of compaction.

Meanwhile, in the process of compression, the maximum value of the force is defined as the peak force (*PF*). *MCF* (meaning crashing force) is the average force during the entire energy dissipation process, which can be defined as the ratio of *EA* to *ECD*.
(13)$$MCF=\frac{EA}{ECD}$$

*CFE* (crushing force efficiency) is defined by the ratio of *MCF* to *PF*, which shows the consistency of the structural load.
(14)$$CFE=\frac{MCF}{PF}$$

The crashworthiness indicators of the AHSTs are summarized in Figure 15. The energy absorption of the double-skin sandwich structure is much higher than that of the single-skin sandwich structure for the same aspect ratio *R*. For example, at an aspect ratio of 1, the initial peak force of 60D60-CHC is 39% higher than that of 60D60-CH, and the *CFE* is also higher. As the aspect ratio *R* of 120D40-CH is 3, it has the lowest initial peak force. The results of the data analysis show that the smaller the aspect ratio of the tube, the more stable the sandwich structure is. The stability of the double-epithelial sandwich structure makes the damage more stable and sufficient, and thus has higher energy absorption.

Comparing previous research work [33], the empty CFRP tubes without filled honeycomb cores were used as a comparison and are summarized in Table 4. The energy absorption of the tube improved after filling the honeycomb. For example, the *SEA* of C1D63 improved from 6.4 J/g to 8.7 J/g, and the *SEA* of C1D40 improved from 8.9 J/g to 9 J/g. The *CFE* of the sandwich tube (43%, 50%, and 67%) was always higher than that of the CFRP thin-walled structure tube (20–25%), which is because the honeycomb stabilized the damage of the sandwich tube compared to the thin-walled tube. In addition, the *PF* of the AHSTs (4.1 kN and 6.6 kN) was significantly higher than that of the CFRP thin-walled tubes (1.5 kN–3.7 kN). The reason is the lightweight material and that the sandwich tube remained a lightweight energy-absorbing structure when this honeycomb was used as filling in the thin-walled structure. 

## 4. Conclusions

In this paper, based on the free-rolling mechanism of the auxetic honeycomb, the honeycomb cylindrical shell was prepared successfully to overcome the fracture problem of the hexagonal honeycomb in the curling process. In addition, the auxetic honeycomb sandwich tubes (AHSTs) with a variable Poisson’s ratio were designed and fabricated by molding and bonding processes. A Poisson’s ratio model for an auxetic honeycomb under uniaxial compression was established, and the relationship between the Poisson’s ratio and reentrant angle was predicted and verified. 

The effects of different geometric parameters and sandwich structures on the structural performance were compared in terms of the failure mode and force–displacement histories. The failure mechanisms of AHST structures were analyzed and their axial mechanical properties were evaluated. Based on this study, the following conclusions can be drawn: The relationship between the Poisson’s ratio and cell topology is studied, and Poisson’s ratios for a series of in-plane cells are defined. The Poisson’s ratio of hexagonal honeycombs is directly affected by the cell topology, and can usually be divided into reentrant, square, and convex honeycombs. Through the topological transformation, the Poisson’s ratio of the cells also changes from negative to positive (i.e., convex cells with positive Poisson’s ratio, zero Poisson’s ratio, negative Poisson’s ratio, and reentrant cells).Through the geometry analysis of the auxetic honeycomb core, the function of the honeycomb reentrant angle along the out-of-plane direction can be obtained. When the honeycomb geometry is constant, the AHST has a minimum curl radius (the inverse of the curvature). As the Poisson’s ratio is related to the reentrant angle of the honeycomb, a honeycomb core with a specific Poisson rate can be obtained by changing the geometry of the honeycomb core.According to the force–displacement curve and the crushing histories, four macroscopic failure modes are proposed. Failure Mode I and II appear when the aspect ratio *R* is relatively small (the value of R is 1 to 2), and the failure process is progressively stably crushing or unstable local buckling. However, the unstable failure of Mode III and IV occurs when the slenderness is relatively large (*R* greater than 3) and shearing and collapsing of the structure are found.By comparing the crashworthiness indicators with those of the CFRP thin-walled tubes and AHSTs, it was found that the honeycomb core can improve the *SEA* and *CFE* of the thin-walled tube, with the improvement in *CFE* both exceeding 100%. There is also a large improvement in peak force (with the same diameter, the *PF* from 3.1 kN to 4.1 kN and 3.7 kN to 6.6 kN), which is the result of the complex failure mechanism (ring mode and Z mode mix mode) of the core and skin in the sandwich tube; this can have more damage mechanisms to absorb energy.

The honeycomb is a lightweight material and the sandwich tube remains a lightweight energy-absorbing structure when this honeycomb is filled in a thin-walled structure. Therefore, it is feasible to use the auxetic honeycomb as a core for sandwich structures and it provides new ideas for the design and preparation of such potential sandwich structures.

## Figures and Tables

**Figure 1 polymers-14-04369-f001:**
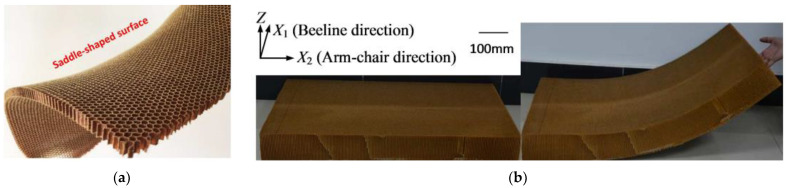
(**a**) Saddle-shaped surface of bent hexagonal honeycomb panel; (**b**) bendable aramid over-expanded honeycomb [14].

**Figure 2 polymers-14-04369-f002:**
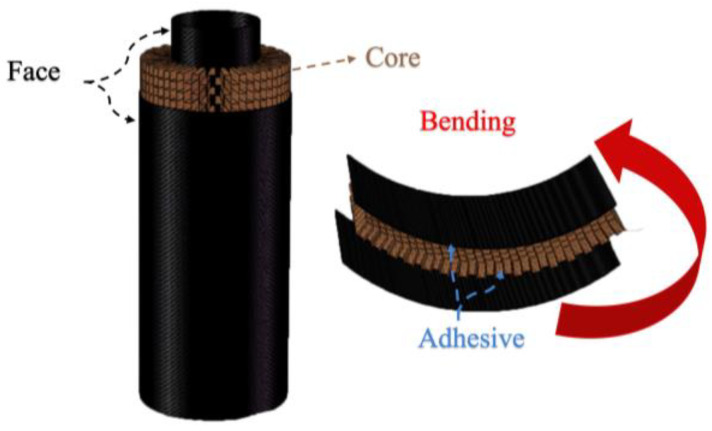
The components of the AHST.

**Figure 3 polymers-14-04369-f003:**
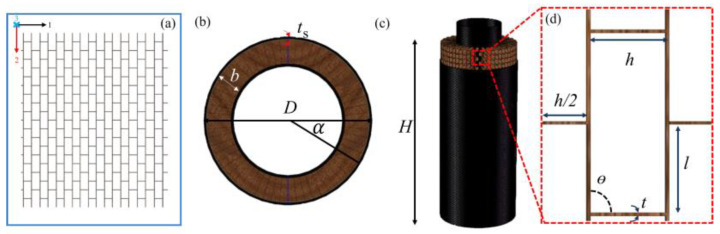
(**a**) Floor plan of the sandwich core, (**b**) vertical view and (**c**) front view of AHST, and (**d**) geometric parameters of the unit cell.

**Figure 4 polymers-14-04369-f004:**
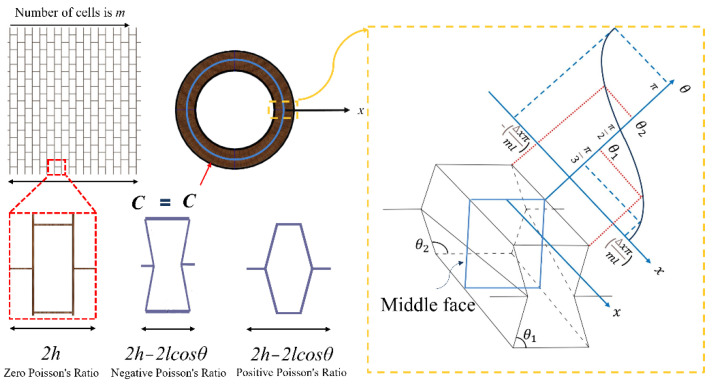
The cell of the core with variable Poisson’s ratio.

**Figure 5 polymers-14-04369-f005:**
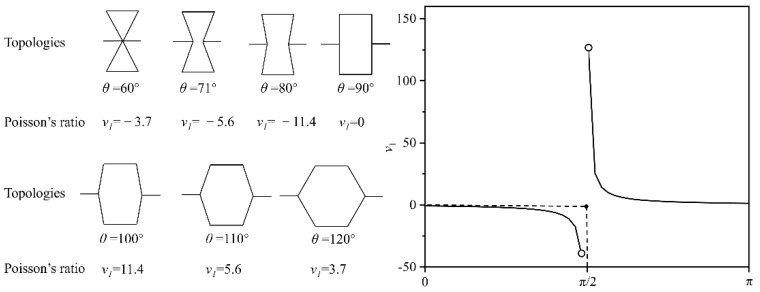
Relationship between topology and Poisson’s ratio.

**Figure 6 polymers-14-04369-f006:**
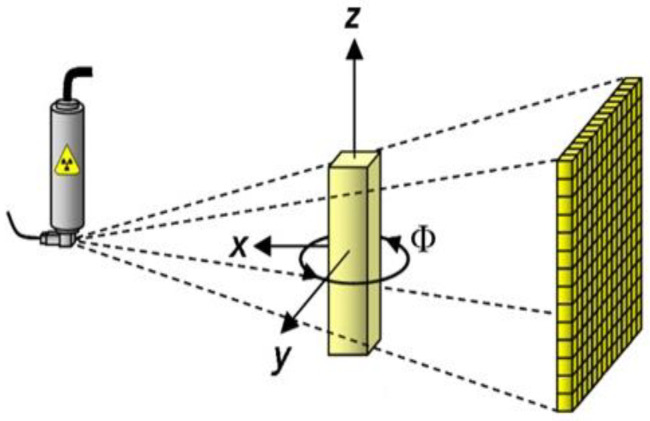
The schematic of the CT scan.

**Figure 7 polymers-14-04369-f007:**
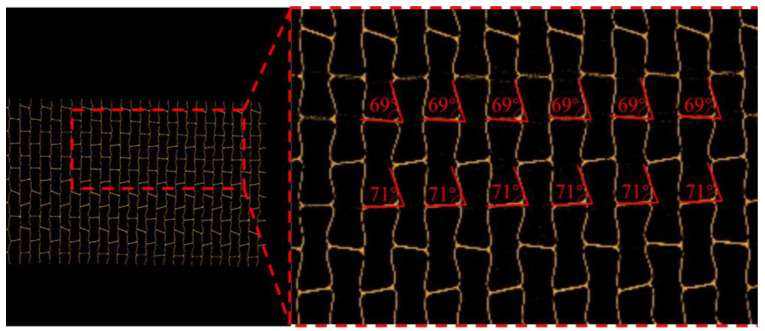
The angles of the deformed honeycomb on the nonplanar expansion of the honeycomb core tube.

**Figure 8 polymers-14-04369-f008:**
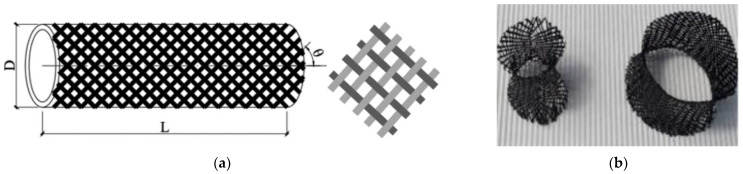
(**a**) Textile structure of braided tubular fabric and (**b**) schematic diagram of the weave of carbon fiber tube [27].

**Figure 9 polymers-14-04369-f009:**
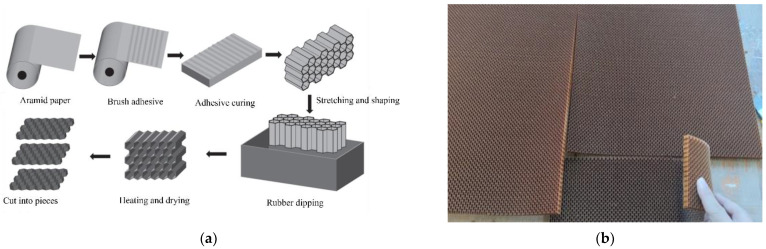
(**a**) The production process of honeycomb [28] and (**b**) bendable aramid auxetic honeycomb.

**Figure 10 polymers-14-04369-f010:**
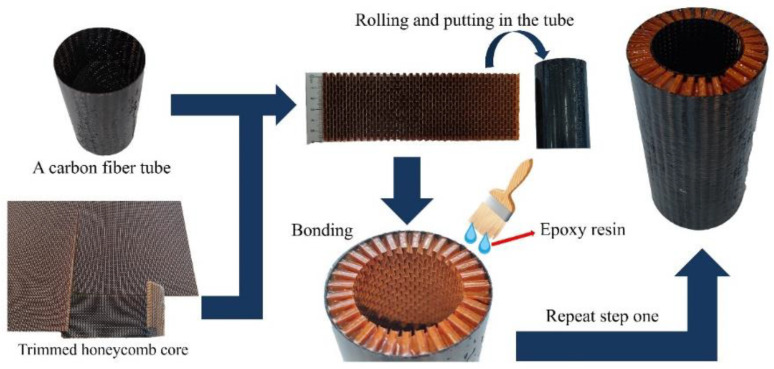
The preparation process and finished product of the AHST.

**Figure 11 polymers-14-04369-f011:**
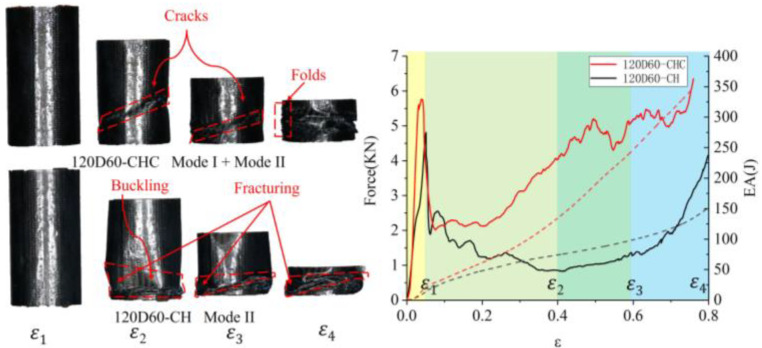
Failure mode and force–displacement curve of 120D60-CHC and 120D60-CH.

**Figure 12 polymers-14-04369-f012:**
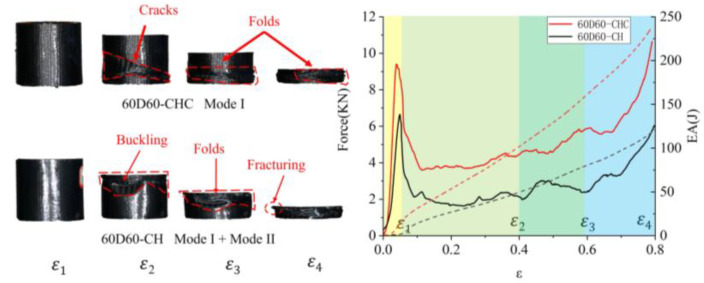
Failure mode and force–displacement curve of 60D60-CHC and 60D60-CH.

**Figure 13 polymers-14-04369-f013:**
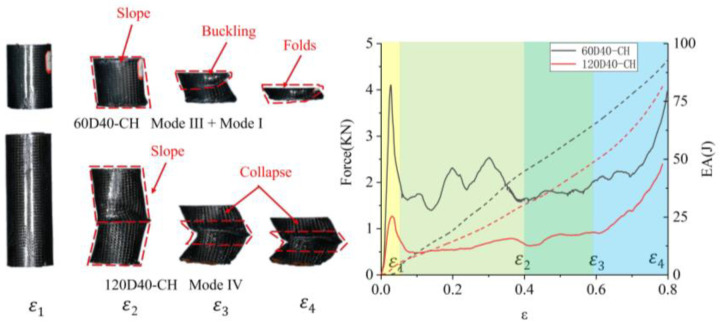
Failure mode and force–displacement curve of 60D4-CH and 120D40-CH.

**Figure 14 polymers-14-04369-f014:**
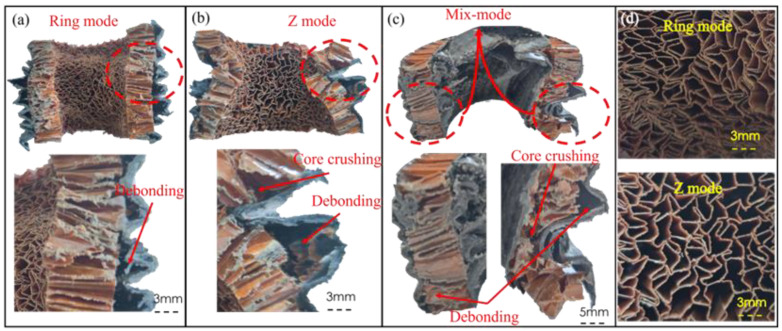
Three failure modes of the thin AHST tube after axial compression. (**a**) the ring mode, (**b**) the Z mode and (**c**) the Mix mode. (**d**) Detail diagram of the failure modes.

**Figure 15 polymers-14-04369-f015:**
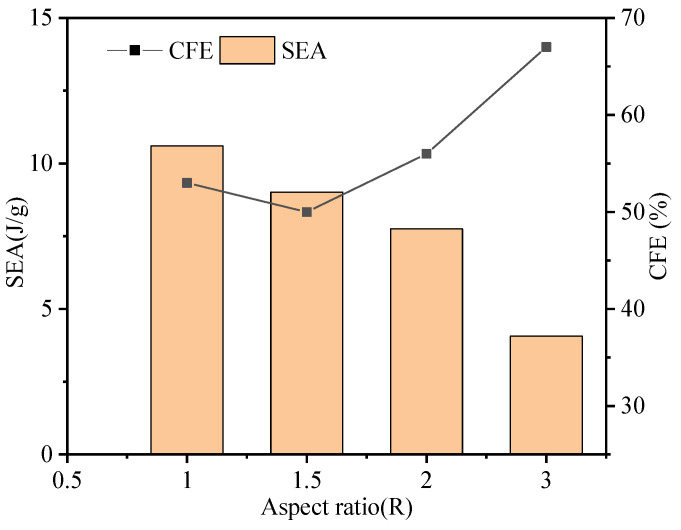
The crashworthiness indicators of the AHSTs with different aspect ratios *R*.

**Table 1 polymers-14-04369-t001:** Material properties of epoxy resin and honeycomb.

Material	Material Property	Value
Epoxy resin	Tension modulus (GPa)	2.69
	Tensile strength (MPa)	57
	Poisson’s ratio	0.3
	Density (g/cm^3^)	0.93
Kevlar honeycomb	Even compression strength (MPa)	2.21
	Longitudinal Shear strength (MPa)	1.28
	Longitudinal Shear modulus (MPa)	117
	Transverse Shear strength (MPa)	0.78
	Transverse Shear modulus (MPa)	35
	Density (g/cm^3^)	0.048

**Table 2 polymers-14-04369-t002:** Summary of the tested specimens.

	*H* (mm)	*b* (mm)	*D* (mm)	*d* (mm)	$t_{s}\;(\mathrm{mm})$	*W* (g)	*R*
60D60-CHC	60	10	60	40	0.5	22.5	1
60D60-CH	60	10	60	40	0.5	15.8	1
120D60-CH	120	10	60	40	0.5	45.0	2
120D60-CHC	120	10	60	40	0.5	34.0	2
60D40-CH	60	10	40	20	0.5	11.0	1.5
120D40-CH	120	10	40	20	0.5	20.0	3

**Table 3 polymers-14-04369-t003:** Summary of crashworthiness indicators and failure modes of specimens.

	*PF* (kN)	*SEA* (J/g)	*EA* (J)	*MCF* (kN)	*CFE* (%)	Failure Mode
60D60-CHC	9.4	10.6	238.56	4.97	53	Mode I
60D60-CH	6.6	8.7	136.74	2.85	43	Mode I + Mode II
120D60-CHC	6.5	7.8	348.84	3.63	56	Mode I + Mode II
120D60-CH	4.8	4.6	158.04	1.65	34	Mode II
60D40-CH	4.1	9	99.11	2.06	50	Mode III + Mode I
120D40-CH	1.3	4.1	81.29	0.85	67	Mode IV

**Table 4 polymers-14-04369-t004:** Summary of crashworthiness indicators of specimens and previous research [33].

	*PF* (kN)	*SEA* (J/g)	*EA* (J)	*MCF* (kN)	*CFE* (%)
C1D32 [33]	1.5	8.0	30.9	0.32	21
C1D40 [33]	3.1	8.9	61.3	0.64	20
C1D50 [33]	1.7	6.1	42	0.44	25
C1D63 [33]	3.7	6.4	73.7	0.77	20
60D60-CH	6.6	8.7	136.7	2.85	43
60D40-CH	4.1	9	99.1	2.06	50
120D40-CH	1.3	4.1	81.3	0.85	67

## Data Availability

The data presented in this study are available on request from the corresponding author.

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
