# Peer review of "An Innovative Auxetic Honeycomb Sandwich Tube: Fabrication and Mechanical Properties"

_polymers, 2022, doi:10.3390/polym14204369_

Round 1
Reviewer 1 Report
N/A
Author Response
Thanks to the reviewers for their suggestions, which helped us a lot with the article. We have carefully revised these suggestions, please see the attachment for details.

Reviewer 2 Report
In this manuscript, the auxetic honeycomb sandwich tube (AHST) with variable Poisson's ratio is manufactured via molding and bonding. The authors prepared the honeycomb cylindrical shell successfully, to overcome the problem of hexagonal honeycomb breakage during curling. The innovative AHST structure made of auxetic aramid fiber honeycomb and carbon fiber reinforced polymer (CFRP) has an excellent mechanical performance. It might be a worthwhile job. However, the manuscript quality is poor. Publishing in the current form is not recommended. Authors should optimize the quality of the manuscript and research content. Here are some comments as follows.
1. Abstract: A good abstract should allow readers to clearly understand the research content of the article and the problem to be solved. At the same time, it is necessary to provide corresponding data in the abstract.
2. Introduction: The introduction requires clear logic, and this part of the manuscript is too rambling. In addition, a comprehensive and concise summary of the current state of research in the corresponding field is required. Moreover, describe the innovative content of the article clearly to increase readability
3. Figure 1: It is difficult to understand the composition and preparation process of AHST from figure 1. A good figure should be carefully designed, including the content and layout, and the logic must be clear. The same question applies to other figures.
4. I am confused by the manuscript's discussion of the findings. The clutter of logic and prose structure makes it difficult for readers to understand the context of their research. There was a need to redesign the writing of the results and discussion sections for clarity. Good writing allows readers to read and understand clearly along the subject.
5. The research content of the manuscript is too thin. It is suggested to increase the research on the structure, morphology, composition, and corresponding physical and chemical properties of the AHST to support the subject and content.
6. Conclusions: Conclusions are not simple statements about the results of an article. It is the further condensing of the research results, which is the sublimation of the results.
Author Response

(The authors gave the same response as above.)

Reviewer 3 Report
Dear Editor
In this study, the fabrication and mechanical properties of the auxetic honeycomb sandwich tube have been investigated.
1. Equations need references
2. Fig.5 (a) can be removed. Add the characterization of the test instrument to the section (2.2 Verification of the honeycomb angle variation)
3. Compare results of this work with other reports
4. Conclusion needs data to show results
5. Please explain more materials of honeycomb
6. Improve the quality of Fig.6
7. Are only tests of failure mode and force-displacement performed?
Author Response

(The authors gave the same response as above.)

Round 2
Reviewer 2 Report
I think the revised manuscript is suitable for the Journal.
Reviewer 3 Report
Dear Editor
The manuscript well has been revised.